# Prevention of Sudden Death Related to Sport: The Science of Basic Life Support—From Theory to Practice

**DOI:** 10.3390/jcm8040556

**Published:** 2019-04-24

**Authors:** Rodrigo Luiz Vancini, Pantelis Theodoros Nikolaidis, Claudio Andre Barbosa de Lira, Cássia Regina Vancini-Campanharo, Ricardo Borges Viana, Marilia dos Santos Andrade, Thomas Rosemann, Beat Knechtle

**Affiliations:** 1Center for Physical Education and Sports, Federal University of Espírito Santo, Vitória 29075810, Brazil; rodrigoluizvancini@gmail.com; 2Exercise Physiology Laboratory, 18450 Nikaia, Greece; pademil@hotmail.com; 3Department of Physical Education, Faculty of Physical Education and Dance, Federal University of Goiás, Goiânia 74690-900, Brazil; andre.claudio@gmail.com (C.A.B.d.L.); vianaricardoborges@hotmail.com (R.B.V.); 4Paulista School of Nursing e São Paulo Hospital, Federal University of São Paulo, São Paulo 04024-002, Brazil; vcassia@hotmail.com; 5Department of Physiology, Federal University of São Paulo, São Paulo 05508-000, Brazil; marilia1707@gmail.com; 6Institute of Primary Care, University of Zurich, 8091 Zurich, Switzerland; thomas.rosemann@usz.ch; 7Medbase St. Gallen Am Vadianplatz, 9001 St. Gallen, Switzerland

**Keywords:** sudden death, basic life support, cardiac arrest, cardiopulmonary resuscitation, physical activity, sport

## Abstract

The sudden cardiac arrest (CA) and death of athletes are dramatic and emotionally impacting events for health professionals, family, and society. Although the practice of sport participation improves general health, physical fitness, and quality of life, intense physical exercise can be a trigger for CA and sudden death occasionally in the presence of known or unknown cardiac disorders (mainly hypertrophic cardiomyopathy) and risk factors (environment, health style, family, and genetic). The present review found that sudden death associated with CA was not such a common event in competitive athletes, but it might be an underestimated event in recreational athletes. Thus, considering the exponential increase in sport participation, both in a recreational or competitive way, and the rate of sudden CA, knowledge of implementing prevention and treatment strategies is crucial. This includes preparation of health professionals and lay people in basic life support (BLS); screening and pre-participation assessment in sport programs and health education; and promotion for the recognition of CA and early completion of BLS and rapid access to automatic external defibrillator to improve the victim survival/prognosis. Thus, the purpose of this review is to provide for health professionals and lay people the most updated information, based on current guidelines, of how to proceed in an emergency situation associated with sudden CA of young adult athletes.

## 1. Introduction

The beneficial role of sports for health has been widely acknowledged; however, sports participation is not without risks, and consequently, the development of safe conditions for athletes should be a major task for practitioners (e.g., coaches, physicians) working with them [1,2]. Sudden death associated with cardiac arrest (CA) represents one of the major challenges for emergency and rehabilitation medicine due to the large number of cases, and their social, economic impact, high morbidity, and mortality [1]. It has been demonstrated [1] in general population that CA Brazilian victims treated for one year in the emergency room presented a high mortality rate (96%) and the occurrence of CA might be associated with other comorbidities such as stress, anxiety, and drugs. The annual incidence of sudden death (related to cardiac disorders) in athletes is lower than in general population. However, when it occurs, it is a cause of consternation and “shocks the general public”, because it is “inadmissible” that elite athletes, who are considered “health models” for the sport spectators, die abruptly. It should be noted that heart disease is the leading cause of sudden death among athletes who are usually asymptomatic victims [2,3]. Overall, the percentage of sudden deaths occurring during or immediately after exercise is approximately 5%, being higher among peoples with 35 years of age or older [2,4]. However, the frequency of sudden death among athletes varies and depends on sample size, sex, training level, type of sport, target population, geographical area, and definition of the condition [5,6,7].

In highly demanding sports in terms of volume and intensity in training and competitions, such as the marathon, triathlon, and ultra-marathon [6,8,9], there is great heterogeneity on the etiology and frequency of sudden death [6,8]. It is important to note that sports with these characteristics of extreme physical effort could negatively impact the health status. Knechtle and Nikolaidis [9] reviewed the physiology and pathophysiology of ultra-marathon running and verified that this type of sport could lead to an increase in creatine-kinase concentration (this enzyme may be a marker of heart damage) and negatively impacts other cardiac biomarkers such as troponin I, electro-, and echocardiograph activities. This scenario could predispose to a sudden death. For example, Strachan et al. [10] showed that an 88-km ultra-marathon could lead to an increase in C-reactive protein, which is associated with heart attack. Moreover, a study on USA triathlon races from 1985 to 2016 reported 135 sudden deaths, resuscitated CA, and trauma-related deaths, whereas the incidence of death or CA among USAT (USA Triathlon) participants was 1.74 per 100,000 [8].

Sudden death in sports has attracted great attention by media (Table 1) showing potential risks of exercise/sport for health [11,12]. Surprisingly, health practitioners exhibited inadequate knowledge of exercise science [11], whereas athletic coaches did not sufficiently meet first-aid standards established by accredited entities [13]. This is a worrying situation since exercise professionals need to provide, in some cases, the initial medical and first aid care on the athletic field in the case of a CA [13]. Thus, the understanding of risk situations of sudden death and CA among athletes can provide information for the creation of prevention and treatment strategies and improvement of the victims’ prognosis [14]. Therefore, this review aims to provide for health and exercise practitioners the more recent theoretical and practical information’s of how to act in an emergency situation associated with sudden death and CA in sports. For the purpose of this study, the term ‘athlete’ refers to those practicing any sport or participating to exercise programs.

## 2. Methodology

This review was performed between 1 September 2018 and 31 November 2018 and was conducted in the following databases and according to specific purposes:

Google’s database—search for emblematic cases of sudden death associated with the practice of sport and countries and localities that promote access to strategies of management and prevention of sudden CA in the context of the school. In our view, school settings are the initial place to start the propagation and consolidation of health promotion and education strategies about first aid and basic life support (BLS). The search strategies used were to cross the terms (sudden death and sport), (first aid/basic life support and school), and (teach and first aid and school).

PubMed—a broad review of the literature was carried out without the restriction of year and with the words: (i) Basic life support, (ii) first aid, (iii) automated external defibrillator (DEA, as the title word and in the abstract), (iv) sudden death, and (v) sport.

In addition, we have used other studies to provide relevant theoretical and practical information regarding to first aid and BLS.

## 3. Results

### 3.1. Sudden Death and Sport: General Aspects and Presumed Causes

Sudden death related to physical exercise and sport can be defined as an unexpected death occurring during or immediately after physical exercise (1–3 h) due to any cause, excluding violence, and may be of cardiac etiology (i.e., hypertrophic cardiomyopathy, anomalous coronary artery disease, arrhythmias, valvular diseases, myocarditis, and coronary atherosclerotic disease) and non-cardiac (i.e., use of illicit drugs, pulmonary embolism, brain diseases such as stroke and hyponatremic encephalopathy, hyperthermia, and rhabdomyolysis) [4,6,15,16,17,18,19,20].

Fortunately, sudden death in athletes is a very rare event (1:50,000–1:100,000 annually) being the result of multifactorial conditions, various disorders, and cardiovascular diseases [17]. Additionally, sudden death is related to hereditary causes, such as structural genetic cardiac conditions (which promote important morphological changes) and arrhythmias (which promote lethal electrical alterations), which are frequently found or suspected to be the cause of death in postmortem examinations in athletes [17,21,22].

In these cases, for individuals with these unknown conditions, intense physical exercise may be the “trigger” for lethal cardiac arrhythmias leading to sudden death [23]. Although regular physical activity increases exercise capacity and plays a role in both primary and secondary prevention of a variety of chronic disorders [24,25], competitive and intense physical exercise is associated with a significant increase of sudden death risk in competitive athletes, especially adolescents and young adults. Several pathogenetic mechanisms have been speculated, including silent cardiovascular conditions, mostly cardiomyopathy, premature coronary artery disease, and congenital coronary anomalies [17,21,22,23]. Unexpected events without the presence of pathological substrates, such as commotio cordis (i.e., cardiac concussion), and abuse of unfair and dangerous doping and performance-enhancing drugs, are also potential causes [18]. However, it is well established that regular physical exercise significantly reduces the risk of death from all causes (including cardiac) as well as sudden death compared to a sedentary lifestyle [12,23].

It should be noted that for athletes older than 35 years, coronary artery disease appears as the main cause of sudden death [2,3]. However, hypertrophic cardiomyopathy is documented as the leading cause of death in young competitive athletes (i.e., aged 35 years or less) and accounts for more than a third of the deaths [6,17,22]. For example, clinical diagnosis of hypertrophic cardiomyopathy is based on a hypertrophied, non-dilated left ventricle (identified by echocardiography and/or magnetic resonance imaging) in the absence of another cardiac, systemic, metabolic, or syndromic disease [17,22]. This disease is multifactorial and characterized by diverse clinical, genetic, and morphologic features, including a risk of sudden death from arrhythmia, diastolic dysfunction, or left ventricular outflow tract obstruction, which is the major determinant of progressive heart failure and sudden CA [17].

Finally, pathophysiological mechanisms of sudden death during physical exercise, and consequently CA, are most of the time, associated with the discharge of catecholamines, which interacts unfavorably with some type of pre-existing arrythmogenic substrate. In addition, intense exercise can lead to dehydration, hyperpyrexia, electrolyte imbalance, and increased platelet aggregation [6,12,17,22]. All these disorders and pathological substrates can lead to an abrupt loss of cardiac mechanic function and consequently to sudden CA and death [6,12]. Table 2 shows the main disturbances and disorders associated with the sudden death of athletes and general population and its respective definitions. Table 3 summarizes the prevalence of causes of sudden cardiac death in young athletes by frequency.

### 3.2. Cardiac Death and Sport: Is It Possible to Prevent It?

Cardiac arrest is defined as the interruption of cardiac mechanical activity in a person expecting restoration of cardiopulmonary and cerebral function and can be considered the greatest emergency and a major public health problem [6,46]. During a CA event, there is circulatory inefficiency and absolute deficiency of tissue oxygenation, with the possibility of irreparable cellular damage in a short time in vital organs such as brain and heart [47]. In North America, CA is considered the leading cause of death, affecting about 350,000 individuals per year [48,49]. In Brazil, approximately 200,000 cases of CA per year are estimated to occur, half of them in an extra-hospital environment [47], which is a factor that may worsen the prognosis of victims for lack of adequate treatment, which includes speed in rescue actions and evolves knowledge in BLS [47,48].

The survival of patients after CA is around 3.4% to 22.0%, with only a small proportion of the victims being discharged alive, and often with persistent neurological damage and poor quality of life [48], as we also observed in a study conducted by our research group [1]. The most common causes of CA can be divided into cardiac (i.e., acute coronary syndromes, cardiac arrhythmias, cardiac valve disease, cardiomyopathies, and congenital heart disease) and non-cardiac causes (i.e., stroke, pulmonary thromboembolism, respiratory failure, airway obstruction, and overdose of drugs and medicines) [50]. The sudden CA can occur in sports and depending on the intensity of physical exertion and the pre-existing disorder, affecting professional and/or recreational athletes [4,6,15,16,17,18,19,20,22]. It should be noted that the interaction between sudden death, CA, and sport has been a focus of much interest by researchers [6,12,23]. Table 4 and Table 5 show the main conclusions related to CA and BLS in the sport context, respectively. In summary, both tables show that cardiac structural and electrical disturbances are important causes of sudden death among athletes. In this context, population education on BLS and early BLS support and defibrillation is fundamental. These interventions could improve the prognosis of survival and neurological recovery of victims.

### 3.3. Strategies for Prevention and Treatment of Sudden Cardiac Death Related to Sport

The implementation of strategies for the prevention and treatment of sudden CA and death, includes team and individual strategies for the management of athletes, associated with public health campaigns, such as universal (health professional and lay people) training in CPR, and the availability of automated external defibrillators (AEDs) in public settings [23]. Despite their importance in the treatment of CA, the number of AEDs in sport facilities is still inadequate to cover the needs of the increased number of athletes [65,66]. In this context, “The Survival Chain” advocated by AHA and BLS techniques may contribute to decrease mortality from heart disease and sudden CA, which may precipitate sudden death events that could be triggered by physical activity and sport.

However, for the promotion and education of first aid to become effective and to be transferred to diverse contexts of society and public and private spaces, including those involving the practice of physical activity and sport, it is necessary that this knowledge and information is transmitted to children and young people in elementary and secondary schools. For instance, this information could be incorporated in the curriculum of physical education course, where a couple of courses annually would cover this topic. In addition, an extra-curriculum seminar could be conducted annually for both teachers and school-children. Table 6 and Table 7 show the studies and their main conclusions related to first aid and AED in the context of sport and physical activity. In addition, Table 8 shows countries that have implemented such policies in the context of the school. In summary, these three tables show that the emergency action plans and campaigns of health promotion and education among lay people and health professionals are essential to improve the outcomes of cardiac events that can lead to CA and consequently to sudden death related to physical activity and sport. However, for these health promotion and education policies and campaigns to be effective, it is necessary to follow the correct guidelines advocated by entities such as the American Heart Association (AHA), such as “The Survival Chain”.

The sudden CA can occur in the context of sports (i.e., professional and/or recreational) and physical activity (i.e., occupational, leisure time, and planned/structured programs) and depend on both the intensity of physical exertion and the pre-existing disorders [4,6,15,16,17,18,19,20,22]. Thus, pre-sports assessment and monitoring to identify the presence of silent heart disease and arrhythmias may reduce the risk of sudden CA related to physical exertion [12]. Sport and physical exercise assessment and monitoring consist of screening for pre-existing cardiac diseases and should include: Anamnesis and physical examination directed to cardiovascular signs/symptoms; and family history of premature cardiac death, familial arrhythmias, and coronary artery disease [12]. In addition, the athletes’ assessment should include other diagnostic exams such as the 12-lead ECG (electrocardiogram) for the investigation of arrhythmias and ischemia and, where appropriate, resting and stress ECG, pharmacological stress test, and imaging tests such as echocardiography and magnetic resonance imaging and catheterization [12].

It should be noted that in Italy, the National Health System recommends the pre-participation screening of all competitive athletes. Long-term Italian experience has provided evidence that pre-participation screening in competitive athletes with 12-lead ECG, history and physical examination is effective in identifying potentially lethal cardiovascular diseases [84]. In the USA, the American College of Cardiology/AHA recommends a pre-participation screening program limited to the use of specific questionnaires and clinical examination [85]. In addition to the pre-participation screening, preparation and training of health professionals and staff involved in the care and follow-up of athletes are essential for the recognition, care, and treatment of a sudden CA [12].

Additionally, the actions and treatment of sudden CA in adults must be based on the “Survival Chain” [86], being an algorithm and an ideal sequence of actions that must be adopted immediately after the recognition of a sudden illness, and consists of interrelated steps that must be followed so that the victims are more likely to survive/prognosis and with the lowest impacts and better possible neurological outcomes [46,87]. Therefore, it is necessary for health professionals, family members and laypersons to know and have training in these actions that must follow an out-of-hospital “Survival Chain” sequence presented in the Figure 1. Considering the increased risk for re-occurrence of CA, secondary prevention and cardiac rehabilitation programs are necessary for those who survived CA [88]. The main components of a subsequent cardiac rehabilitation program are presented in Figure 2.

The BLS is a set of measures that aim to maintain blood flow to vital organs. Early recognition of sudden CA and emergency medical system (EMS) activation (currently, this is easier as we are in the “era of smartphones”), early CPR, and rapid defibrillation are key to increasing survival and improving the prognosis of victims [87]. According to the American Heart Association [29,82], the steps to perform BLS are:
Assess whether the location is safe in order to ensure there is no physical risk for the rescuer and the victim;Evaluate if the victim is aware by touching your shoulders and asking out loud, “Are you okay?” and/or “Can you hear me?”;Ask for help, call the EMS, and request an AED;Check at the same time whether the victim has pulse and normal breathing;Observe for effective movement of the chest and palpate the carotid pulse for a maximum of 10 s;If the patient has a pulse and breathes normally, monitor the patient until the EMS arrives;If the victim has a pulse but abnormal breathing, initiate rescue breaths by administering one ventilation every 6 s. In this situation, reassess the victim every two minutes, until the EMS arrives;In the absence of pulse, initiate high-quality CPR immediately, intercalating 30 external chest compressions (ECC) cycles with two ventilations, until the arrival of the EMS and AED (Figure 3).

With the arrival of AED, immediately stop what you are doing and according to standard procedures, position it on the victim. Do this, after opening the case and pressing the power button. The function of the AED is to proceed with the heart rhythm analysis, checking if it is a shockable rhythm or not. If it indicates a shockable rhythm, apply the shock and restart ECC immediately after. If the heart rhythm is not shockable after AED analysis, restart the ECC immediately [29,86,87]. According to Kleinman et al. [87] and the American Heart Association [86], the steps for performing high-quality CPR are as follows:
Put the victim in the supine position and on a flat and rigid surface;Place the hypothenar region of the dominant hand in the lower half of the sternum of the victim and the other hand parallel to the first;Keep the elbows extended, forming an angle of 90° in a horizontal plane;Compress the chest to the depth of 5 to 6 cm in a rhythm (frequency—trained in conjunction with metronome—of 100 to 120 compressions/minute, allowing the total return of the thorax at each compression;After 30 ECC, perform two ventilations;For ventilation to be effective, open the airways before applying them and observe the elevation of the chest at each ventilation;The ventilations should be fast, lasting approximately 1 s and performed with the bag-valve-mask device coupled to the oxygen source (when possible);Relieve the rescuer who does the ECC every two minutes, thus avoiding their fatigue and decreasing the quality of the CPR.

As a key part of BLS, defibrillation is most effective when applied early. The AED is a device capable of analyzing the heart rhythm, identifying the need for shock, and emitting an electric shock in order to return the heart to sinus rhythm [87]. However, before placing the AED, make sure that the patient is unconscious, without effective breathing, and without a pulse. Follow the instructions below [86,87], which are the universal steps for using the AED:
Place the adhesive pads on the patient’s chest—below the right clavicle and at the apex from heart;Attach the blade connector to the unit;Evaluate the heart rhythm. During the analysis of the rhythm and application of the shock, make sure that you and others are away from the victim;After the shock, immediately start CPR for two minutes.

It is necessary to emphasize that some conditions can interfere in the analysis of the rhythm and the application of the shock. In case of excessive hairs on the thorax, trim them; if the victim’s chest is wet, dry the site; in patients with pacemakers, apply the blades approximately 2 to 2.5 cm from the protuberance on the skin and when there is a medication patch on the victim’s chest, remove it and position the blades [86,87]. BLS interventions should be maintained until the arrival of the EMS at the emergency site, so that advanced life support measures such as maintenance of an advanced airway and drug administration can be performed [87].

For children, many similarities exist compared to adults; the main differences between children and adults are [89]:
For children, if two rescuers are available to do CPR, the compression to breaths ratio is 15:2; if only one rescuer is available, the ratio is 30:2 for all age groups.For very small children, you can use one-handed chest compressions.The depth of compression may be different. For a child, compress the chest at least one-third the depth of the chest. This may be less than two inches for small children but will be approximately two inches for larger children.If you are the only person at the scene and find an unresponsive child, perform CPR for two minutes BEFORE you call EMS or go and look for an AED.In children, primary cardiac events are not common. Cardiac arrest is most commonly preceded by respiratory problems. Survival rates improve with early intervention for respiratory problems. Remember that prevention is the first link in the pediatric chain of survival!If you witness a cardiac arrest in a child, call EMS and get an AED just as you would in the adult BLS sequence

The key question is how long CPR and BLS should be performed. Regarding CPR and BLS in athletes in the field, it must be continued until ALS (advanced life support) has arrived and the athlete will be transferred to a specialized unit where ALS will continue.

A limitation of the present review was that it focused on sports settings; thus, caution would be needed to generalize its findings to other settings, e.g., occupation. A further limitation is the fact that these studies had small sample sizes and most studies were performed with men. Therefore, the knowledge for women is small.

On the other hand, strength of this study was its novelty, since—to the best of our knowledge—it was the first one to review this topic. Considering the increased practice of exercise, reflected in the increased number of annual outdoors endurance races, such as half-marathons and full marathons [90,91], the findings would be of great practical importance for practitioners working with endurance athletes. It should be noted that this increased number of races stressed the need for availability of first aids services. In addition, it should be highlighted that the majority of these athletes were recreational athletes who started regular exercise at an adult age. 

Furthermore, the risk for sudden CA as well as for any other health implication during exercise should not discourage humans from their participation in these physical activities. Instead, the risks for health during exercise should be considered in the context of weighting the risk-to-benefit ratio in these activities [92]; since exercise has a beneficial role for health, humans should be encouraged to participate in such activities despite potential risks.

## 4. Conclusions

Sudden CA due to sport participation at the presence of risk factors (in general cardiac disorders) may have known or unknown etiology; the seemingly healthy and safe practice can become the “trigger” for this tragic and dramatic event for athletes, relatives, spectators, and society in general. It is a “shocking and devastating” event, as it often involves the unexpected death of a young healthy person. In this context, pre-participation screening, primary prevention (cardiovascular risk assessment), and secondary strategies (training of health professionals and laypeople to perform BLS, i.e., identification of sudden CA, EMS activation, application of CPR, and early use of AED) increase chances of survival and improve the prognosis of the victims. In this way, health professionals who work directly or indirectly with athletes should be trained and seek the theoretical and technical preparation to act in emergency situations in the correct way, which may be decisive for the survival of the victims. In this sense, access to information and periodic training courses are essential and should be a mandatory part of the undergraduate curriculum and promoted, publicized, and sponsored by institutions (public and private) in the field of exercise and sport. In our view, communication of this information should be initiated in elementary and secondary schools, initially in a playful way and later in a more systematic way (e.g., courses) in accordance with the most current guidelines. Moreover, the findings of the present review highlight the need for an increased availability of AEDs in sport facilities. In summary, a combination of prevention and treatment strategies would be expected to decrease health risks—and particularly, the occurrence of sudden CA—in athletes.

## Figures and Tables

**Figure 1 jcm-08-00556-f001:**
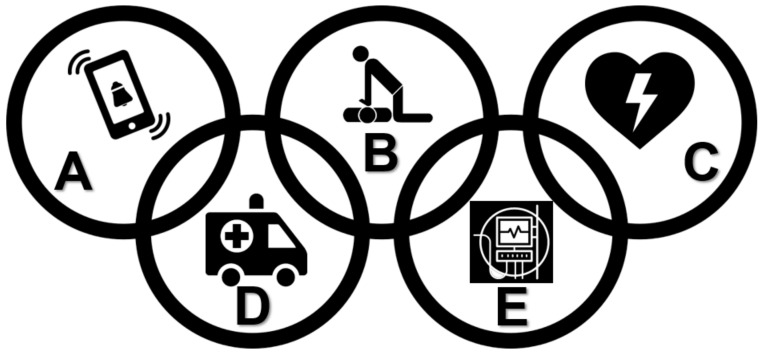
Out-of-hospital “Chain of Survival” of Sport (adapted from AHA [86]). (**A**) Recognition and activation of the emergency medical system, which varies according to country, city and locality; (**B**) immediate high-quality cardiopulmonary resuscitation, which includes administration/training/education in BLS; (**C**) rapid defibrillation—fast access to the automated external defibrillator; (**D**) access to advanced medical emergency services, which includes on-site care for sudden CA and transport to the in-hospital environment. (**E**) Advanced life support and post-cardiopulmonary resuscitation care—all of these actions occur in the hospital setting.

**Figure 2 jcm-08-00556-f002:**
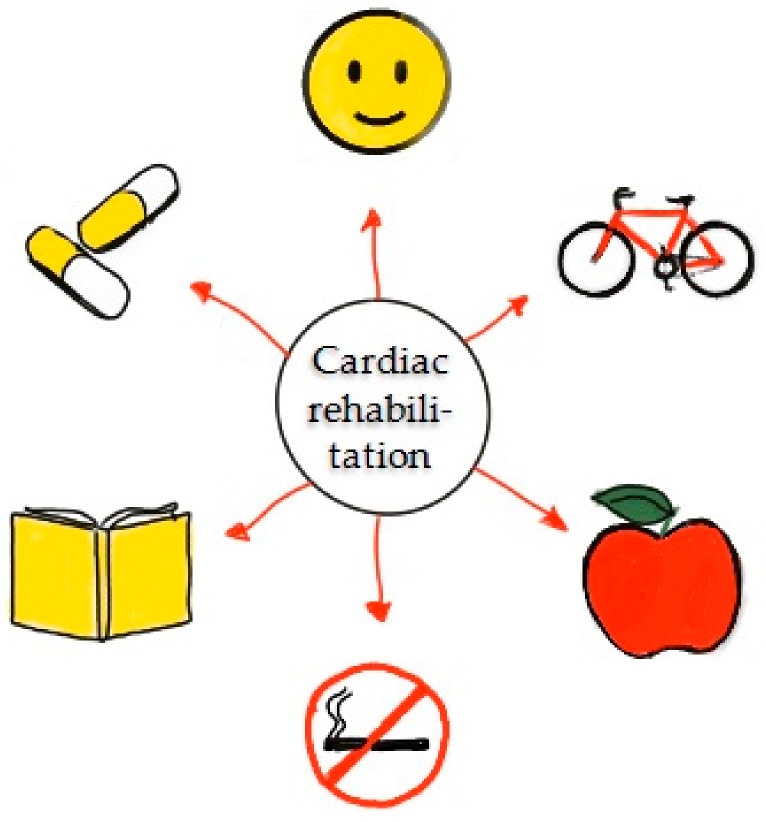
Components of a cardiac rehabilitation program. Such a program includes interventions on behavior, exercise, nutrition, smoking cessation, education, and medicines [88].

**Figure 3 jcm-08-00556-f003:**
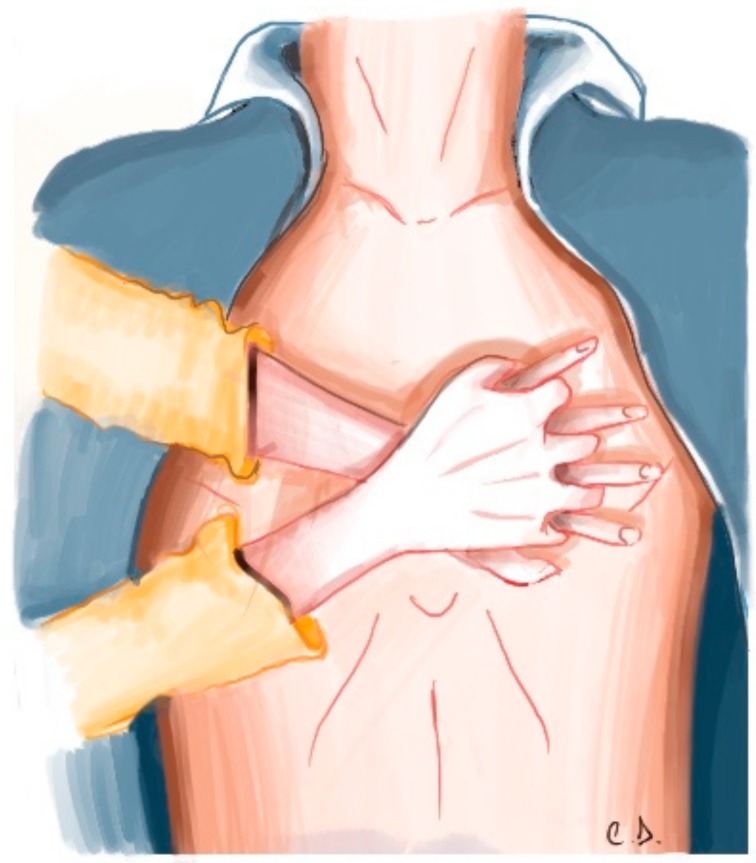
Location of chest compression.

**Table 1 jcm-08-00556-t001:** News reporting sudden death associated to the sport practice (*n* = 16) with great media repercussion and cause of death (Google’s database).

Athlete	Gender	Sport	Age (Years)	Reported Cause	Link
1	Male	Soccer	31	Bradyarrhythmia	http://www.espn.com/soccer/fiorentina/story/3408662/davide-astori-died-natural-death-from-heart-issue-autopsy-shows
2	Male	Basketball	26	Natural	http://www.espn.com/mens-college-basketball/story/_/id/18668751/former-syracuse-boston-celtics-center-fab-melo-dies-brazil
3	Female	Biathlon	21	Acute heart failure	https://www.reuters.com/article/us-biathlon-russia-yakimkina-idUSKBN0LR0Z720150223
4	Male	Soccer	25	SCA	http://www.espn.com/soccer/italian-serie-b/story/2950188/doctors-found-guilty-of-manslaughter-over-death-of-piermario-morosini
5	Male	Soccer	22	SCA	https://www.theguardian.com/football/2007/aug/28/europeanfootball.sevilla
6	Male	Soccer	30	Heart attack	http://news.bbc.co.uk/2/hi/americas/3967127.stm
7	Male	Soccer	24	Heart attack	https://www.theguardian.com/football/2004/jan/26/newsstory.sport4
8	Male	Soccer	28	Heart attack	https://www.theguardian.com/world/2003/jun/26/football
9	Male	Swimming	26	Atherosclerotic coronary artery disease	https://www.reuters.com/article/us-swimming-oen/norwegian-swim-champ-dale-oen-died-of-heart-disease-official-idUSBRE85B10820120612
10	Male	Volleyball	37	Heart attack	https://www.foxsports.com/olympics/story/vigor-bovolenta-dies-age-37-italian-volleyball-player-olympic-silver-medalist-heart-attack-032512
11	Female	Volleyball	26	Thrombosis	http://www.fivb.org/viewPressRelease.asp?No=34247&Language=en#.W65-fmhKjDc
12	Male	Soccer	26	Heart failure	https://www.theguardian.com/football/2009/aug/09/espanyol-dani-jarque-dies
13	Male	Soccer	35	Heart failure	https://www.theguardian.com/football/2007/dec/30/newsstory.sport2
14	Male	Soccer	26	Heart failure	https://www.bbc.com/sport/football/36675570
15	Male	Soccer	25	Heart collapse	http://edition.cnn.com/2010/SPORT/football/03/09/football.africa.deaths/index.html
16	Male	Soccer	27	Heart failure	https://www.theguardian.com/football/2013/aug/01/christian-benitez-died-heart-failure

SCA: sudden cardiac arrest; the Google database was searched combining the words “sudden death” and “sport” (in the link news and all) since 2003. The access date on the abovementioned websites was 31 November 2018.

**Table 2 jcm-08-00556-t002:** Main disturbances and disorders associated with the sudden death of athletes and general population.

Cause	Literal Definition	Reference
Cardiomyopathies	Defined by structural and functional abnormalities of the ventricular myocardium those are unexplained by flow-limiting coronary artery disease or abnormal loading conditions.	[26]
Hypertrophic cardiomyopathy	Defined by the presence of increased left ventricular wall thickness that is not solely explained by abnormal loading conditions.	[27]
Coronary artery anomalies	Can be classified as anomalies with obligatory ischemia, without ischemia or with exceptional ischemia. The clinical symptoms may include chest pain, dyspnea, palpitations, syncope, cardiomyopathy, and arrhythmia.	[28]
Arrhythmia (the main is ventricular fibrillation)	Electrical malfunction in the heart that causes an irregular heartbeat.	[29]
Valvulopathies	Changes in the heart valves, which cause stenosis or insufficiency, and may result in hemodynamic problems.	[30]
Myocarditis	Defined as inflammation of the heart muscle that may be identified by clinical or histopathologic criteria and an important cause of dilated cardiomyopathy. Viral infection is an important cause.	[31]
Coronary atherosclerotic disease	Develops when the major blood vessels that supply heart with blood, oxygen, and nutrients (coronary arteries) become damaged or diseased. Cholesterol-containing deposits in arteries (plaque, accumulate at the site of injury in a process called atherosclerosis) and inflammation are usually to blame for disease.	[32]
Heart attack	Occurs when a blocked artery prevents oxygen-rich blood from reaching a section of the heart. If the blocked artery is not reopened quickly, the part of the heart normally nourished by that artery begins to die. The longer a person goes without treatment, the greater the damage.	[29]
Cardiac arrest	Occurs suddenly and often without warning. It is triggered by an electrical malfunction in the heart. With its pumping action disrupted, the heart cannot pump blood to the brain, lungs, and other organs. Seconds later, a person loses consciousness and has no pulse. Death occurs within minutes if the victim does not receive treatment.	[29]
Pulmonary embolism	Characterized by occlusion of one or more pulmonary arteries. Physiological disturbance may be minimal, but often cardiac output decreases as the right ventricle attempts to overcome increased afterload. Additionally, ventilation-perfusion mismatches can develop in affected vascular beds, reducing systemic oxygenation.	[33]
Stroke	Occurs when the blood supply to part of your brain is interrupted or reduced, depriving brain tissue of oxygen and nutrients. Within minutes, brain cells begin to die.	[34]
Hyponatremic encephalopathy	Symptomatic cerebral edema due to a low osmolar state. Hyponatremia is a common electrolyte disturbance occurring in a broad spectrum of patients, from asymptomatic to critically ill. The disease is defined as a decrease in serum sodium concentration to a level below 136 mmol per liter. The brain damage from hyponatremia can be associated with either hyponatremic encephalopathy or improper therapy of symptomatic hyponatremia.	[35,36]
Hyperthermia	May be caused by excessive heat production, diminished heat dissipation, or malfunction of the hypothalamic thermostat.	[37]
Rhabdomyolysis	Defined as injury of the skeletal muscle, which results in the release of intracellular contents into the circulation. Skeletal muscle comprises 40% of body mass, and a large insult can result in the accumulation of cellular contents in the extracellular space such that elimination mechanisms are overwhelmed. The resulting effects are recognized as a clinical syndrome of muscle injury that is associated with the development of myoglobinuria, electrolyte abnormalities, and often acute kidney injury	[38]
Respiratory failure	Condition in which not enough oxygen passes from your lungs into your blood. When respiratory failure causes a low level of oxygen in the blood, it is called hypoxemic respiratory failure. When respiratory failure causes a high level of carbon dioxide in the blood, it is called hypercapnic respiratory failure.	[39]
Pulseless electrical activity	Also known as electromechanical dissociation is a clinical condition characterized by unresponsiveness and impalpable pulse in the presence of sufficient electrical discharge. Electrical activity is a pertinent, but not sufficient, condition for contraction. Also defined as organized ECG activity, excluding ventricular tachycardia and fibrillation, without clinical evidence of a palpable pulse or myocardial contractions	[40,41]

AHA: American Heart Association; NHLBI: National Heart, Lung, and Blood Institute.

**Table 3 jcm-08-00556-t003:** Summary of the prevalence of causes of sudden cardiac death in young athletes by frequency.

Coronary artery anomaly	1:100
Bicuspid aortic valve	1:100
Hypertrophic cardiomyopathy	1:500
Wolff-Parkinson-White	1:750
Short QT syndrome	1:2000
Arrhythmogenic right ventricular cardiomyopathy	1:2000 to 1:5000
Brugada syndrome	1:2000 to 1:5000
Long QT syndrome	1:2500
Dilated cardiomyopathy	1:2500
Marfan syndrome	1:5000
Catecholaminergic polymorphic ventricular tachycardia	1:10,000

Data from different sources [42,43,44,45].

**Table 4 jcm-08-00556-t004:** Studies and their main conclusions related to cardiac arrest (CA) in the sport context.

Author	Year	Study Type	Title	Journal	Country	Main Findings and Conclusions
Prech et al. [51]	2018	Case report	An amateur marathon runner after sudden cardiac arrest. Whether and when can he return to sport competition?	Polski Merkuriusz Lekarski	Poland	A case of an amateur runner has been presented, competitor in 50 marathons, who underwent a sudden CA. Reports on the possible dangers of overdosing extreme endurance exercises are necessary.
Landry et al. [52]	2017	Original	Sudden cardiac arrest during participation in competitive sports	The New England Journal of Medicine	Canada	Incidence of sudden CA during participation in competitive sports was 0.76 cases per 100,000 athlete-years within a specific region of Canada, with 43.8% of the athletes surviving until they were discharged from the hospital. Two deaths were attributed to hypertrophic cardiomyopathy. The occurrence of sudden CA due to structural heart disease was uncommon during participation in competitive sports
Zorzi et al. [53]	2016	Original	Sudden cardiac arrest in Italian sports facilities in 2015: epidemiological implications of the so-called “Balduzzi decree”	Giornale Italiano di Cardiologia (Rome)	Italy	After the sudden cardiac death of elite athletes in 2012, the presence of an AED and professionals trained to perform CPR must be available in Italian sports facilities. In 2015, it was reported 123 cases of sudden CA in Italian sports facilities. The majority victims were males (93%) and >35 years old (88%). The rate of return of spontaneous circulation was 62% when an AED was used before electromyostimulation arrival vs. 9% when no bystander CPR or AED use by lay rescuers was mentioned.
Harmon et al. [54]	2016	Original	Incidence and etiology of sudden cardiac arrest and death in high school athletes in the United States.	Mayo Clinic Proceedings	USA	The rate of sudden CA and death in male high school athletes was 1:44,832 athletes per year, with almost half due to possible or confirmed cardiomyopathy disease.
Solberg et al. [55]	2016	Original	Sudden cardiac arrest in sports—need for uniform registration: A position paper from the Sport Cardiology Section of the European Association for cardiovascular prevention and rehabilitation	European Journal of Preventive Cardiology	European position	Rational decisions about cardiac pre-participation screening and cardiac safety at sport facilities requires increased data quality concerning incidence, etiology, and management of sudden CA/sudden death in sports.
Cronin et al. [56]	2013	Original	Prepared for sudden cardiac arrest? A cross-sectional study of automated external defibrillators in amateur sport	British Journal of Sports Medicine	Ireland	A total of 81.3% of amateur clubs owns an AED. Many clubs engage in regular maintenance and storage of AEDs. However, this study identifies several areas for improvement in facilitating a secure survival chain for players in the event of a sudden CA.
Marijon et al. [57]	2013	Original	Characteristics and outcomes of sudden cardiac arrest during sports in women	Circulation: Arrhythmia and Electrophysiology	France	Compared with men, the incidence of sudden cardiac death in women was lower. Despite similar circumstances of occurrence, survival at hospital admission (46.5%) was significantly higher than that for men (30.0%). Favorable neurological outcomes were similar (80%). Cause of death seemed less likely to be associated with structural heart disease in women compared with men (58.3% vs. 95.8%).
Enright et al. [58]	2012	Original	Primary cardiac arrest following sport or exertion in children presenting to an emergency department: chest compressions and early defibrillation can save lives, but is intravenous epinephrine always appropriate?	Pediatric Emergency Care	Australia	The importance of early CPR and defibrillation in collapsed young athletes (with cardiac disorders that suffer CA) is fundamental. These interventions could result in full long-term neurological recovery.
Kramer et al. [59]	2010	Review	Review of the management of sudden cardiac arrest on the football field	British Journal of Sports Medicine	South Africa	Trained medical professionals must be allowed to respond, ideally with a defibrillator, to a player who suddenly and unexpectedly collapses on the field. Immediate defibrillation of a pulseless ventricular tachycardia or ventricular fibrillation has a successful cardioversion rate exceeding 90%. Medical professionals should be well trained and rehearsed in the recognition of sudden CA. Prompt initiation of CPR together with early defibrillation, will result in many athletes’ lives being saved.
Cappato et al. [60]	2010	Original	J wave, QRS slurring, and ST elevation in athletes with cardiac arrest in the absence of heart disease: marker of risk or innocent bystander?	Circulation: Arrhythmia and Electrophysiology	Italy	J wave and/or QRS slurring was found more frequently among athletes with CA/sudden death than in control athletes. Nevertheless, the presence of this ECG pattern appears not to confer a higher risk for recurrent malignant ventricular arrhythmias.
Furlanello et al. [61]	1998	Original	Cardiac arrest and sudden death in competitive athletes with ARVD	Pacing and Clinical Electrophysiology	Italy	ARVD is a predisposing factor for sport-related CA. Prevalence of ARVD among athletes with CA is high, confirming the observation that ARVD is one of the major causes of sudden death. All CA were athletic activity related, indicating the potentiality of exercise as a cause of electrical destabilization in subjects with ARVD.
Maron et al. [62]	1995	Original	Blunt impact to the chest leading to sudden death from cardiac arrest during sports activities	The New England Journal of Medicine	USA	Sudden death from CA in a young person may occur during sports play after a blunt blow to the chest in the absence of structural cardiovascular disease or traumatic injury (commotio cordis). The authors identified 25 cases (people of 3–19 years) from the registries. Incidents took place during organized competitive sports in 16 cases and in recreational settings in 9. Twelve victims collapsed virtually instantaneously on impact, whereas 13 remained conscious and physically active for a brief time before CA. CPR was administered within about 3 min to 19 victims, but normal cardiac rhythm could be restored in only 2. The authors concluded that most sudden deaths related to impact to the chest are due to ventricular dysrhythmia induced by an abrupt, blunt precordial blow, presumably delivered at an electrically vulnerable phase of ventricular excitability.
Peters and Reil [63]	1995	Original	Risk factors of cardiac arrest in arrhythmogenic right ventricular dysplasia	European Heart Journal	Germany	Arrhythmogenic right ventricular dysplasia is an important cause of ventricular arrhythmia with a potential risk of sudden cardiac death. People with structural alterations and a low level of right ventricular function is at a high risk of CA and strenuous exercise and sport remain most important risk factors.
Dietlen et al. [64]	1953	Not available	Acute cardiac arrest in sport exercise and work	Münchner Medizinische Wochenschrift	Germany	Not available

CA: Cardiac arrest; AED: Automated external defibrillator; CPR: Cardiopulmonary resuscitation; ARVD: Arrhythmogenic right ventricular dysplasia.

**Table 5 jcm-08-00556-t005:** Studies and their main conclusions related to basic life support (BLS) in sports.

Author	Year	Study Type	Title	Journal	Country	Main Findings and Conclusions
Karam et al. [65]	2017	Original	Major regional differences in Automated External Defibrillator placement and Basic Life Support training in France: Further needs for coordinated implementation	Resuscitation	France	There is great heterogeneity in public AEDs programs. Population education on BLS offers an important benefit, regardless of the density of AEDs implanted, which should be taken into account when planning public health policies to improve the survival of sudden CA outside the hospital setting.
Gonzalez et al. [66]	2009	Case report	Ventricular fibrillation during sport activity successfully treated	Brazilian Archives of Cardiology	Brazil	Public access programs to the AED may increase survival from ventricular fibrillation in the out-of-hospital setting. It is necessary to stimulate the training of lay people for the use of AED and BLS and to disseminate these behaviors in places with great circulation of people and those with high risk of sudden death (sports centers and arenas).
Mills et al. [67]	1997	Review	The athlete’s heart	Clinics in Sports Medicine	USA	Coaches and other sports professionals should learn BLS measures such as CPR. Such efforts may prevent heart deaths related to physical exercise.

AED: Automated External defibrillator; CA: Cardiac arrest; CPR: Cardiopulmonary resuscitation; BLS: Basic life support.

**Table 6 jcm-08-00556-t006:** Studies and their main conclusions and results related to first aid in the context of sport and physical activity.

Author	Year	Study Type	Title	Journal	Country	Main Findings and Conclusions
Schneider et al. [68]	2017	Original	Health care in high school athletics in West Virginia	Rural Remote Health	USA	In order to avoid a potentially fatal emergency or possibly sudden cardiac death, emergency planning should be an essential part of high school sports programs. The requirement for first aid, CPR certification, and an emergency action plan are steps that can improve the health care of athletes from rural areas and non-rural areas.
Lear et al. [69]	2015	Original	Preventing sudden cardiac death: automated external defibrillators in Ohio high schools	Journal of Athletic Training	USA	Emergency action plans in secondary schools should emphasize the management of sudden CA in sports facilities and the placement of AEDs. Over a period of 11 years, 25 episodes of AED use were recorded in 22 secondary schools, of which 20 occurred in or near sports facilities.
Kramer et al. [70]	2015	Review	F-MARC: promoting the prevention and management of sudden cardiac arrest in football	British Journal of Sports Medicine	Switzerland	To prevent and administer sudden CA in soccer, the FIFA Medical and Research Center is establishing a program of research, education, and practical standardization. This strategy detected players at medical risk during mandatory pre-competitive assessments. In addition, FIFA disseminated accepted guidelines for the interpretation of athletes’ electrocardiogram, developed field-specific protocols for recognition, response, CPR, and removal of a player who underwent sudden CA.
Marijon et al. [71]	2011	Original	Sports-related sudden death in the general population	Circulation	France	Sudden sport-related death in the general population is more common than previously thought. Most cases are witnessed, but CPR by “lay” viewers was initiated in only one third of cases. Immediate interventions were significantly associated with improved survival and prognosis.
Brion [72]	2010	Review	Sport-related sudden death and its prevention	Bulletin de L’AcadémieNationale de Médecine	France	The immediate causes of sudden sports-related death are age dependent. Prevention begins with screening. Before 35 years, the most frequent causes are hypertrophic cardiomyopathy and arithmogenic right ventricular cardiomyopathy. People with cardiovascular diseases at risk of sudden death should adapt their sports activities. Knowledge of first aid procedures by those who supervise sports activities may improve prognosis.
Rich [73]	1994	Review	Sudden death screening	Medical Clinics of North America	USA	Emergency plans need to be established by physicians and sports coaches. The recognition of cardiac symptoms associated with sudden CA is a key point. Appropriate education, which includes information and training in first aid, BLS and CPR, should be encouraged for those working with athletes. The emergency plan should include on-site treatment, the way to contact Emergency medical services, and transportation to a qualified health care facility.
Kassanoff et al. [74]	1972	Original	Stadium coronary care. A concept in emergency health care delivery	JAMA	USA	BLS and emergency stations were established at sports stadiums as part of a research program to elucidate the precipitating mechanisms and/or events of sudden CA and thereby suggest appropriate therapeutic intervention through a well-equipped and prepared emergency team. During the study period there were 20 episodes of myocardial infarction. It was verified the occurrence of 1 acute event in most sports events with participation of 30,000 to 40,000 people.

CPR: Cardiopulmonary resuscitation; CA: Cardiac arrest; AED: Automated external defibrillator; BLS: Basic life support.

**Table 7 jcm-08-00556-t007:** Studies and their main conclusions related to automated external defibrillator in the context of sport and physical activity.

Author	Year	Study Type	Title	Journal	Country	Main Findings and Conclusions
Kinoshi et al. [75]	2018	Correspondence	Mobile automated external defibrillator response system during road races	The New England Journal of Medicine	Japan	Between 2005 and 2017, of 1,965,265 runners in 251 road races (of 10.0 to 42.2 km), 30 CA were attended. Shocks were delivered to 23 runners who had ventricular fibrillation, and another 5 runners (4 with pulseless electrical activity and 1 with ventricular fibrillation) recovered with basic CPR only. The median interval between collapse and the return of spontaneous circulation was 5.5 min. All these runners had return of spontaneous circulation in the field and had a favorable neurologic outcome.
Fortington et al. [76]	2017	Original	“It doesn’t make sense for us not to have one”—understanding reasons why community sports organizations chose to participate in a funded automated external defibrillator program	Clinical Journal of Sport Medicine	Australia	Implementation of AEDs in community sports settings is an important component of emergency medical planning. Two overarching themes emerged: Awareness of the program and decision to apply.
Luther et al. [77]	2016	Original	A collapsed sportsman with a shock advised in sinus rhythm: the importance of automated external defibrillator rhythm strip retrieval prior to defibrillator implantation.	Circulation: Arrhythmia and Electrophysiology	England	AEDs rely on rhythm detection algorithms with high specificity for recognizing ventricular arrhythmia when used appropriately. AED diagnostics can be difficult to retrieve once the patient has arrived in hospital, with patient management decisions often made in their absence.
Kramer [78]	2013	Editorial	Automated external defibrillator in sport: absolutely always available	British Journal of Sports Medicine	South Africa	Sudden CA remains the leading cause of death in sports. Whenever its prevention has failed, for whatever reason, its immediate medical management becomes paramount if the life under acute threat is to be saved. In many circumstances, this can only be effectively and efficiently achieved by the presence of a fully functional AED which is activated while CPR is being undertaken. It is no longer a question of whether an AED is necessary in any mass gathering sport environment but how many are necessary and where they should be located
Toresdahl et al. [79]	2013	Original	High school automated external defibrillator programs as markers of emergency preparedness for sudden cardiac arrest	Journal of Athletic Training	USA	A total of 2784 schools (82.6%) reported having 1 or more AEDs on campus, with an average of 2.8 AEDs per school; 587 schools (17.4%) had no AEDs. Schools with an enrollment of more than 500 students were more likely to have an AED. Suburban schools were more likely to have an AED than were rural, urban, or inner-city schools. Schools with 1 or more AEDs were more likely to ensure access to early defibrillation, establish an emergency action plan for sudden CA, review the emergency action plan at least annually, consult EMS to develop the emergency action plan, and establish a communication system to activate emergency responders. High schools with AED programs were more likely to establish a comprehensive emergency response plan for sudden CA.
Smith and Hoogenboom [80]	2011	Review	The use of cardiopulmonary resuscitation and the automated external defibrillator in the practice of sports physical therapy	International Journal of Sports Physical Therapy	USA	During the initial assessment of the injured athlete, the sports physical therapist must first be concerned with life-threatening emergencies such as absence of breathing and pulse. The sports physical therapist must also be aware of the possibility of sudden CA that could occur in others, including coaches, officials, and fans. Therefore, skills and ongoing certification in CPR techniques and the use of an AED are a basic necessity. These skills are required as part of the specialty practice of sports physical therapist and are mandatory for being qualified.
Drezner et al. [81]	2011	Original	Automated external defibrillator use at NCAA Division II and III universities	British Journal of Sports Medicine	USA	81% of institutions had at least one AED in the university athletic setting. Athletic training rooms (75%) were the most likely location to place an AED. Twelve cases of AED use for sudden CA were reported with 67% occurring in older non-students, 16% in intercollegiate athletes, and 16% in students (non-intercollegiate athletes). The AED deployed a shock in eight cases. Eight of 12 (66%) victims were immediately resuscitated, but only 4 survived to hospital discharge (overall survival 33%). None of the intercollegiate athletes or students survived. Although no benefit was demonstrated in a small number of intercollegiate athletes, AEDs were successfully used in older individuals on campus with CA.
Ngai et al. [82]	2010	Case report	A patient with commotio cordis successfully resuscitated by bystander cardiopulmonary resuscitation and automated external defibrillator	Hong Kong Medical Journal	Hong Kong	Sudden deaths of young people during competitive sports are usually due to congenital heart diseases. Ventricular fibrillation, however, may also occur in individuals with no underlying cardiac disease who have sustained a low-impact chest wall blow (commotio cordis). Successful resuscitation can be achieved by prompt CPR and early defibrillation. Accessible CPR-trained personnel and AEDs should be present at all organized sporting events.
Coris et al. [83]	2005	Original	Sudden cardiac death in division I collegiate athletics: analysis of automated external defibrillator utilization in National Collegiate Athletic Association division I athletic programs	Clinical Journal of Sport Medicine	USA	Sixteen departments that previously reported having had a sudden cardiac death event at their institution responded to this follow-up telephone survey. 20% of AED uses were attributed to student athletes, with 33% of utilizations for athletic department staff and 47% for fans. Defibrillation was actually administered in 53% of AED unit applications. Time to shock was an average of 3.4 min, with average EMS response time of 8.2 min for those events without EMS on site. Reported survival to hospital discharge in this university athletic department setting for sudden cardiac death was 0% for students, 75% for staff, 57% for fans, and 61% overall. Athletic department AED programs were extremely successfully at increasing survival of sudden cardiac death far above national prehospital standards, mainly in the nonathletic population.

CA: Cardiac arrest; CPR: Cardiopulmonary resuscitation; AED: Automated external defibrillator; EMS: Emergency medical services.

**Table 8 jcm-08-00556-t008:** Resources about promotion and education of first aid initiatives in different countries in the school context (Google’s database).

	Short Title	Link
England	Plan to teach all children first aid	https://www.bbc.com/news/education-44883708
Australia	First aid in schools program	https://www.stjohnvic.com.au/first-aid-in-schools.asp
Ireland	First aid for schools	https://pulse8.ie/first-aid-for-schools/
USA	Teaching CPR in middle schools and high schools	https://aha.channing-bete.com/school.html
France	Teach first aid to children younger than 6 years	https://bmjopen.bmj.com/content/bmjopen/4/9/e005848.full.pdf
Scotland	Call for every school to teach pupils CPR and first aid	https://www.sundaypost.com/news/scottish-news/call-every-school-teach-pupils-cpr-first-aid/
Israel	Teach first aid to Bedouin high school students	https://msih.bgu.ac.il/volunteer-opportunities-in-medical-school/
Canada	School programs	http://www.calgary.ca/CSPS/Recreation/Pages/Teacher-resources/First-Aid-grades-7-to-12.aspx
Brazil	Social project: mass training	http://www.socesp2015.com.br/treinamentoemmassa/

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
