# Peer review of "Prevention of Sudden Death Related to Sport: The Science of Basic Life Support—From Theory to Practice"

_jcm, 2019, doi:10.3390/jcm8040556_

Round 1

Reviewer 1 Report

Thank you for the opportunity to review this review paper of Prevention of sudden death related to sport: The science of basic life support - From theory to practice. Editorial/ grammar/ sentence structure along with writing fluency revisions are needed throughout the paper.

Introduction

-          Lack of a perspective more integrative. Please try to shorten the introduction.

-          Check other cardiovascular risk factors that may affect to CA and sudden death. E.g. stress and anxiety? Drugs?

-          Please provide a summary of how many marathon and so on have taken place during the last year. It would be nice to highlight that the number of those events are increasing, so it would help to highlight the need of knowing how to do BLS and using an AED among the general population…

Results

-          Please, add a title ‘results’ before the following section: 2.1. Sudden death and sport: general aspects and presumed causes…

-          Please add some information about the lack of AED. There are some studies where it has been reported that there are mor fire extinguishers than AED

-          Table 2. Main disturbances and disorders associated with the sudden death of athletes and general population. Please, specify the % of prevalence of every disorder associated with the sudden death according to the bibliography.

-          “However, for the promotion and education of first aid to become effective and to be transferred 203 to diverse contexts of society and public and private spaces, including those involving the practice of 204 physical activity and sport; it is necessary that this knowledge and information are transmitted to 205 children and young people in elementary and secondary schools” Please, add some comments about how frequently BLS should be explained to children and young adults. Every year?

-          Please, add some comment that BLS in children is different. Maybe it would have to be explained with BLS for adults.

-          Please, consider to add a picture explaining the place you have to compress the chest.

-          Please, consider to add that the rhythm of compressions is the same rhythm than ‘ Staying alive’ and other songs.

-          Limitations of the population of the studies to generalization of the results should be addressed (low % of women, sample size,...) I guess most of the studies have been done among men. Is it right?

-          Please, make some comments about how long the BLS should be done… Please, add comments about neurological issues after BLS too long…

-          Please, add some comments about the relevance of secondary prevention and cardiac rehabilitation programs on those who survived after a CA.

- Please in the figures, consider to add a picture related to cardiac rehabilitation because it would be part of the survival chain

Conclusion

-          Please, re-write the discussion in order to make it more insightful

-          Please, provide more information about the need of better services when marathons,…

-          Please, provide a general interpretation of results in context to prior evidence.

-          Please, highlight the benefits of physical activity and exercise among the population. We don’t want to scare people from practicing exercise.

-          Please, add information about the need of more AED available.

Author Response

Thank you for the opportunity to review this review paper of Prevention of sudden death related to sport: The science of basic life support - From theory to practice. Editorial/ grammar/ sentence structure along with writing fluency revisions are needed throughout the paper.

Answer: We agree with the expert reviewer and edited all text carefully for writing fluency. 

Introduction

-          Lack of a perspective more integrative. Please try to shorten the introduction.

Answer: We agree with the expert reviewer; we developed an integrative perspective (in the beginning of the introduction: „The beneficial role of sports for health has been widely acknowledged; however, sports participation is not without risks, and consequently, the development of safe conditions for athletes should be a major task for practitioners (e.g. coaches, physicians) working with them [1,2].“) and shortened the introduction.

 -          Check other cardiovascular risk factors that may affect to CA and sudden death. E.g. stress and anxiety? Drugs?

Answer: We agree with the expert reviewer and added this aspect in the introduction („and the occurrence of CA might be associated with other comorbidities such as stress, anxiety and drugs.“).

 -          Please provide a summary of how many marathon and so on have taken place during the last year. It would be nice to highlight that the number of those events are increasing, so it would help to highlight the need of knowing how to do BLS and using an AED among the general population…

Answer: We agree with the expert reviewer and developed this aspect in the practical applications before conclusions.

 Results

Please, add a title ‘results’ before the following section: 2.1. Sudden death and sport: general aspects and presumed causes…

Answer: We agree with the expert reviewer and added a heading “Results” and changed all subsequent subheadings accordingly.

 Please add some information about the lack of AED. There are some studies where it has been reported that there are more fire extinguishers than AED

Answer: We agree with the expert reviewer and added this aspect („Despite their importance in the treatment of CA, the number of AEDs in sport facilities is still inadequate to cover the needs of the increased number of athletes [61,62].)“.

 Table 2. Main disturbances and disorders associated with the sudden death of athletes and general population. Please, specify the % of prevalence of every disorder associated with the sudden death according to the bibliography.

Answer: We agree with the expert reviewer and added a new table showing the prevalence of the most important etiologies.

  “However, for the promotion and education of first aid to become effective and to be transferred 203 to diverse contexts of society and public and private spaces, including those involving the practice of 204 physical activity and sport; it is necessary that this knowledge and information are transmitted to 205 children and young people in elementary and secondary schools” Please, add some comments about how frequently BLS should be explained to children and young adults. Every year?

Answer: We agree with the expert reviewer and added this information („For instance, this information could be incorporated in the curriculum of Physical Education course, where a couple of courses annually would cover this topic. In addition, an extra-curriculum seminar could be conducted annually for both teachers and school-children.“).

 Please, add some comment that BLS in children is different. Maybe it would have to be explained with BLS for adults.

Answer: We agree with the expert reviewer and added the steps in BLS in children.

 Please, consider to add a picture explaining the place you have to compress the chest.

Answer: We agree with the expert reviewer and added figure 3 showing this information.

 Please, consider to add that the rhythm of compressions is the same rhythm than ‘ Staying alive’ and other songs.

Answer: We agree with the expert reviewer and

 Limitations of the population of the studies to generalization of the results should be addressed (low % of women, sample size,...) I guess most of the studies have been done among men. Is it right?

Answer: We agree with the expert reviewer and added this aspect in the limitations

 Please, make some comments about how long the BLS should be done… Please, add comments about neurological issues after BLS too long…

Answer: We agree with the expert reviewer and added ‘The key question is for how long CPR and BLS should be performed. Regarding CPR and BLS in athletes in the field, it must be continued until ALS (advanced life support) has arrived and the athlete will be transferred to a specialized unit where ALS will continue’

 Please, add some comments about the relevance of secondary prevention and cardiac rehabilitation programs on those who survived after a CA.

Answer: We agree with the expert reviewer and added such information and a detailed figure.

Please in the figures; consider adding a picture related to cardiac rehabilitation because it would be part of the survival chain

Answer: We agree with the expert reviewer and added a figure on cardiac rehabilitation.

 Conclusion

Please, re-write the discussion in order to make it more insightful

Answer: We agree with the expert reviewer and revised the discussion significantly.

 Please, provide more information about the need of better services when marathons,…

Answer: We agree with the expert reviewer and added this aspect in the paragraph before conclusions.

 Please, provide a general interpretation of results in context to prior evidence.

Answer: We agree with the expert reviewer and added a general interpretation („In summary, a combination of prevention and treatment strategies would be expected to decrease health risks - and particularly, the occurrence of sudden CA - in athletes.“).

 Please, highlight the benefits of physical activity and exercise among the population. We don’t want to scare people from practicing exercise.

Answer: We agree with the expert reviewer and developed the idea of risk-to-benefit ratio in the paragraph before conclusions.

 Please, add information about the need of more AED available.

Answer: We agree with the expert reviewer and added this aspect in the conclusions („Moreover, the findings of the present review highlight the need for an increased availability of AEDs in sport facilities.“).

Reviewer 2 Report

 The purpose of this review was to provide for health professionals and lay people the most updated information, based on current guidelines, of how to proceed in an emergency situation associated with sudden CA of young adult practitioners of physical activity and sport. This review is very important to know for several populations. So I recommend to publish this after revision.

1) Could you add or showt for accessed day in the references in Table 1?

2) Is there any study and  limitation of this review article?

Author Response

 The purpose of this review was to provide for health professionals and lay people the most updated information, based on current guidelines, of how to proceed in an emergency situation associated with sudden CA of young adult practitioners of physical activity and sport. This review is very important to know for several populations. So I recommend to publish this after revision.

1) Could you add or show for accessed day in the references in Table 1?

Answer: We agree with the expert reviewer and added the date of access to references below the Table 1.

2) Is there any study and limitation of this review article?

Answer: We agree with the expert reviewer and added the limitations of this study.